# The Efficacy and Safety of a Personalized Protocol Designed to Balance Hemoglobin Levels in Hemodialysis Patients as Led by Nephrology Clinical Nurse Specialists: An Intervention Study

**DOI:** 10.3390/healthcare13111317

**Published:** 2025-06-02

**Authors:** Ruth Israeli, Gillie Gabay, Sigal Shafran Tikva, Michal Exman, Irit Mor Yosef Levi, Ruth Radiano, Rely Alon, Yulia Lerman, Revital Zelker

**Affiliations:** 1Dialysis Unit, “Ziv”, Hadassah Ein-Kerem Medical Center, Jerusalem 91120, Israel; ruthy@hadassah.org.il (R.I.); daks54@hadassah.org.il (M.E.); iritmy@hadassah.org.il (I.M.Y.L.); 2School of Science, Achva Academic College, Shikmim 79804, Israel; gillie.gabay@gmail.com; 3Nursing Administration, Hadassah Ein-Kerem Medical Center, Jerusalem 91120, Israel; tsigal@hadassah.org.il (S.S.T.);; 4Nursing Administration, Hadassah Mount Scopus University Hospital, Jerusalem 9765422, Israel; radiano@hadassah.org.il; 5Nursing and Health Professions, Hadassah Medical Center, Jerusalem 91120, Israel; relya@hadassah.org.il

**Keywords:** anemia, clinical nurse specialist, dialysis, end-stage renal disease, hemoglobin balance, nephrology, safety, empowerment

## Abstract

**Background:** The ongoing clinical challenge of managing hemoglobin levels in chronic dialysis patients is exacerbated by the gap between growing patient needs and the limited availability of nephrologists. Clinical nurse specialists (CNSs) have been contributing to the successes of modern healthcare systems across Europe, but there is a limited understanding of specific mechanisms by which CNSs can support and improve patient outcomes in renal diseases. **Objectives:** Responding to previous calls, this intervention study evaluated the role of nephrology CNSs in dialysis patient care. **Methods:** This intervention study employed erythropoiesis-stimulating agents (ESAs) to investigate whether a personalized, tailored protocol led by nephrology CNSs could improve the hemoglobin balance compared to the conventional standard of care led by nephrologists. Thirty-nine patients who met the inclusion criteria with a preset hemoglobin value between 10.5–12 g/dL completed the study. **Results:** There were no significant differences in hemoglobin levels between patients managed by nephrologists and those overseen by CNSs. Hemoglobin variability remained unchanged after protocol implementation, while key dialysis quality indicators (e.g., iron saturation, urea reduction) remained within safety limits. Notably, ESA-related medical adjustments were significantly reduced, requiring half as many modifications over 12 study points. **Conclusions:** A CNS-led personalized protocol effectively maintained dialysis patient parameters within established safety thresholds. These findings reinforce the critical role of CNSs in enhancing the efficiency, effectiveness, and safety of hemoglobin management in this high-risk population. Policy implications are discussed.

## 1. Introduction

Anemia, which is a common complication of chronic kidney disease (CKD), is characterized by low levels of the iron-rich protein hemoglobin (HgB), which transports oxygen in red blood cells [1,2,3]. Guidelines define anemia levels in CKD as below 13.0 g/dL or 12.0 g/dL for males and females, respectively, versus ranges of 13.5–17.5 g/dL and 12.0–15.5 g/dL for non-anemia–CKD healthy males and females [4,5]. The main causes of anemia among CKD patients are a deficiency of erythropoietin (EPO), a hormone required for red blood cell (RBC) production [6,7], chronic inflammation, poor gastrointestinal iron absorption, and blood loss [8]. The overall prevalence of anemia in CKD is estimated to be 15.4%, with the frequency increasing as renal disease advances [9]. Debilitating symptoms caused by the anemia include fatigue, reduced productivity, cognitive impairment, shortness of breath, decreased exercise tolerance, dizziness, headaches, loss of appetite, and depression [10,11,12,13]. Anemia in CKD is associated with a higher prevalence or risk of cardiovascular disease, significantly increases the risk of cardiovascular morbidity, congestive heart failure, kidney failure, hospitalizations, and general mortality [8,9]. All these symptoms may significantly decrease the quality of life of CKD patients [14,15,16].

The impact of anemia on the lives of CKD patients highlights the importance of optimal anemia management. In the advanced stages of kidney disease, the condition is associated with increased hospitalization rates, higher mortality, and a reduced quality of life [17]; however, CKD and its associated symptoms may be undertreated among those who do not depend on dialysis [7,12]. This may be a mistake, since low HgB levels may accelerate the progression of CDK, thereby exposing patients to elevated risks of cardiovascular morbidity and mortality [7,12,17]. The anemia that is common in end-stage renal disease (ESRD) patients who receive hemodialysis is typically due to a lack of erythropoietin [12,18,19], whose replacement raises persistent challenges in the balancing of end-stage CDK patients [2,20]. Notably, EPO replacement therapy often results in fluctuating HgB levels, which represent a well-documented clinical challenge for nephrologists in balancing anemia [19,21]. Consequently, balancing anemia in ESRD requires informed coordination between physicians, nurse practitioners, physician assistants, registered nurses, dietitians, pharmacists, social workers, and case managers [21]. As kidney failure progresses, the erythropoietin deficiency becomes more pronounced [21], and variability in HgB levels may be exacerbated by standard treatment regimens that fail to account for endogenous delays in response to therapy [19]. The resulting HgB instability makes it challenging to achieve and maintain values within the recommended target range [22]. In the absence of a single diagnostic test to determine iron status and toxicity risks, providers must adopt a comprehensive approach to iron management, ensuring sufficient iron replenishment to overcome hepcidin blockade, while mitigating the risks associated with excessive iron exposure [22,23].

### 1.1. ESA Therapy

Since its approval in 1989, ESA therapy has been the primary treatment approach for anemia in patients with chronic renal failure [24]. Initially, the treatment target was set to achieve HgB levels comparable to those of healthy adults, which necessitated high ESA doses. However, accumulating evidence of the adverse effects associated with high-dose ESA therapy has led to a reassessment of treatment targets [25]. The use of erythropoiesis-stimulating agents (ESAs) to manage anemia in ESRD patients presents additional complexities and requires careful clinical oversight, because although ESA therapy may reduce mortality, achieving target HgB levels too fast can be dangerous. Excessively high HgB levels (>12 g/dL) have been linked to an increased risk of cerebral infarction, cardiac events, and mortality [22]. The lack of a universally accepted treatment goal for ESA use complicates care [19,24,25] and emphasize the need for a personalized rather than a standardized approach to ESA dosing. A randomized controlled trial (RCT) involving 62 hemodialysis patients followed for over one-year demonstrated that individualized ESA dosing reduced the HgB variability compared to a population-based protocol [26].

A calibrated model, based on simulations of historical HgB levels and patient responses to prior EPO doses, was utilized to develop new dosing strategies to stabilize HgB levels [19]. The most recent guidelines for improving kidney disease outcomes recommend a target HgB range of 10–11.5 g/dL, with an emphasis on customized treatment [27,28], particularly when managing HgB fluctuations triggered by surgery, infections, or gastrointestinal bleeding in hemodialysis patients [29].

Protocols require weekly monitoring of the increase in Hgb levels following ESA treatment until stability is reached. A reasonable target is an increase of 1 g/dL in HgB levels within the first month of treatment. If the increase in HgB levels is excessive (>1 g/dL over 2 weeks), the ESA dose should be reduced by 25% to 50% [8]. It is also necessary to monitor the blood pressure, which may increase during treatment in some patients. The risk benefit balance must be calculated for each CKD patient, where the benefits of ESA treatment are avoiding blood transfusions and improvement in anemia-related symptoms, while the risks are the toxicity and adverse events associated with higher doses of ESA [8]. Thus, EPO replacement therapy is complex and may produce oscillating Hgb levels; EPO dosing protocols do not account for endogenous delays and appear to contribute to or cause variability in Hgb levels [19].

Studies have demonstrated the benefits of integrated care for anemia management in ESRD patients [30,31]. Studies that assessed the contribution of nephrology nurses to anemia management have reported increased compliance with guidelines among nurses as compared to other health professionals, and a reduction in the number of patient visits to emergency departments over a year [31,32]. The implementation of the Gerrish Act [33] now allows nephrology nurses to prescribe medications to stabilize chronic dialysis patients, including balancing anemia, which benefits patients through the empowerment of nurses to make decisions regarding their care.

### 1.2. Nephrology CNS

There is increasing global interest in expanding the scope of nursing practice by introducing advanced specialized roles [34]. One recognized category of advanced specialized nursing is the clinical nurse specialist (CNS) [35]. According to the International Council of Nurses (ICN), CNSs provide expert-level clinical care within specialized clinical practice areas based on established diagnoses [35]. In some countries, the CNS role is integrated into healthcare systems, encompassing diverse responsibilities across various settings [36], including oncology and palliative care, burns and plastics, hematology, renal disease, rheumatology, and urology [34,37,38]. Within these divisions, CNSs primarily focus on clinical duties [34,36,37]. Beyond their clinical responsibilities, CNSs contribute to quality care by promoting patient-centered care, leading nurse-led clinics, assessing holistic patient needs, advocating for patients, facilitating support groups, and addressing the psychological concerns of both patients and their families [37]. Importantly, CNSs have been shown to reduce healthcare costs [37,39,40].

In this context, the CNS role is widely recognized as essential to the success of contemporary European health systems, which face financial constraints, austerity measures, and workforce shortages [34]. CNSs are valued for their expertise in enhancing care quality, improving patient outcomes, and increasing system efficiency [36,41]. Despite these acknowledgments, there remains a lack of comprehensive understanding regarding how CNSs contribute to improved patient outcomes [37] and further assessment will be necessary to evaluate the effectiveness and contributions of the CNS role across additional specialties [35,36,37,38]. Such evaluations will be critical for optimizing resource allocation within healthcare systems. In response to this need, this intervention study examined the role of the nephrology CNS in the context of kidney failure.

At the end of 2022, 91.9% of Israeli dialysis patients (n = 6962) were treated with hemodialysis [42]. An analysis of nurse–patient and physician–patient ratios revealed a ratio of 7.3 patients per nephrology CNS, compared to 32.5 patients per nephrology physician [43]. This disparity highlights the substantial burden of care placed on specialist physicians, and motivates the exploration of alternative models of care, including a personalized, dedicated nursing protocol within clearly defined boundaries. Previous studies have demonstrated that nurses who adhered to the guidelines of the National Kidney Foundation Disease Outcomes Quality Initiative (KDOQI) effectively reduced complications, emergency department referrals, and hospital readmissions [44,45]. In addition, care delivered by designated nurses was associated with improvements in ESRD outcomes [46]. The current intervention study was designed to evaluate the implementation of a personalized protocol for managing anemia in dialysis patients within a tertiary academic medical center, led by nephrology CNSs. The study was designed to evaluate whether the application of a CSN-led protocol could improve HgB stability and overall anemia management in this patient population.

## 2. Methods

### 2.1. Previous Methods Used to Preserve HgB Stability in a Dialysis Unit

Prior to this study, the regulation of HgB levels in dialysis patients in the medical center focused on in this study was exclusively the responsibility of a designated nephrologist. Although specific baseline data were unavailable, nurses reported significant challenges in maintaining stable HgB levels. These challenges were primarily attributed to the limited availability of nephrologists, despite the existing requirements for weekly assessments and ad hoc therapeutic decisions. Concerns regarding this approach prompted the development of a personalized protocol that transferred specific decision-making authority from nephrologists to nephrology CNSs within predefined safety limits. Given their greater availability and direct patient interactions, a CNS-led personalized care protocol was hypothesized to be able to stabilize HgB levels as effectively as a nephrologist-led management or even potentially improve patient outcomes.

### 2.2. Sample

The inclusion criteria were as follows: chronic ambulatory hemodialysis patients; attendance of the nephrology unit for at least 3 months; diagnosis of clinically stable stage 4 renal disease. Exclusion criteria were as follows: concurrent hematologic disease; hospitalization during the previous 2 weeks; presence of fever, chills, or bleeding. The patients’ HgB value was required to be between 10.5 and 12 g/dL, with changes in monthly values not exceeding ±1 g/dL (no extreme volatility, without decreases in HgB values above 1 g/dL in one week). Patients undergoing hemodialysis who maintain HgB levels within our specific target range (10.5–12 g/dL) constitute a relatively rare subgroup; this is not only the case within our institution but also in broader clinical settings, since levels typically fluctuate significantly.

As mentioned in previous studies [47,48,49], there was a significant challenge in identifying, recruiting, and retaining hemodialysis patients for the study. Of a total of 90 patients who received treatment during the study period, only 39 met the inclusion criteria and completed the full 18-month follow-up. Sixteen (41.03%) patients dropped out during the data collection process, due to their underlying comorbidity, and none of these patients returned in the following months.

Only one patient exhibited significant HgB fluctuations (from 12.6 to 10.3 g/dL) after protocol implementation, and this was thought to be linked to abscess drainage during the data collection period. Table 1 presents the breakdown of dropouts by dialysis period, referring to the number of patients for whom any data were collected in each period of the study. In a few cases, specific data were not collected due to the patients suffering severe health conditions.

A total of 39 ESRD patients on hemodialysis with the relevant narrow range of HgB values completed the intervention protocol. This is considered valuable information within the clinical context, since this specialized patient population has previously posed recruitment challenges [47,48,49].

### 2.3. Recruitment of Nurses, Qualifications, and Implementation Process

Nephrology CNSs involved in HgB regulation for dialysis patients were required to complete a structured training program. This program included daily departmental training consisting of theoretical lectures, real case scenario analyses, practical exercises, and a final examination. Participation was limited to CNSs working only in dialysis units. Prior to training, CNSs were required to review the protocol thoroughly and formally acknowledge their understanding by signing into the institutional procedural portal. Upon successful treatment of five supervised patients, CNSs received authorization to use the protocol autonomously. The inclusion criteria for nurses were as follows: registered nurses; completion of an advanced nephrology course and mandatory training (lectures, examination, and hands-on experience); written authorization obtained from nursing management.

The structured training program included 6 months of protocol implementation under supervision. Prior to this implementation phase, the program involved four modules: procedures and legal aspects, provided by the institutional protocol and procedure coordinator; medical aspects of anemia causes and treatment and EPO use, provided by a nephrologist and a clinical pharmacist; information about the protocol, provided by the head nurses of the nephrology unit and by supervisors; and practical application through group simulations. These modules were followed by a final examination designed to assess comprehension.

The modules were developed collaboratively by expert nurses with extensive experience in nephrology, in partnership with an academic nurse consultant, the institutional protocol and procedure coordinator, and the professional development unit, under the managerial supervision of the head manager of the nursing administration. The training program was overseen by the professional development unit, which authorized nurses to manage HgB levels upon successful completion of the program.

### 2.4. Protocol Description

A group of nephrology CNSs conducted a comprehensive review of the up-to-date literature before designing the personalized CNS-led protocol. The protocol underwent expert nephrologist review and received institutional and ethical approval. Figure 1 outlines the protocol milestones, beginning with a monthly blood count. Protocol activation was initiated when a nephrologist identified eligible patients based on predefined criteria (Appendix A). The nephrologist determined the type and dosage of ESA and the target hemoglobin range. Subsequently, a medical order for the protocol was issued.

Each patient was assigned to a nephrologist and an authorized nephrology CNS, who assumed responsibility for subsequent ESA treatment adjustments based on current and previous blood test results, within the designated safety thresholds. Safety measures stipulated that CNSs could only intervene when HgB values ranged between 10.5 and 12 g/dL and when changes from the previous month’s values did not exceed ±1 g/dL. CNSs were not authorized to alter the type of ESA. For HgB levels within the designated range, CNSs were permitted to adjust the ESA dosage by ±25%. Notably, CNSs were required to report HgB values outside the 10.5–12 g/dL range immediately to a nephrologist and to conduct repeat testing at the next dialysis session.

### 2.5. Study Setting

The study was conducted between 2016 and 2018 in an Israeli tertiary academic medical center. The ambulatory dialysis unit provides specialized care for dialysis patients.

### 2.6. Study Design

This was a prospective comparative intervention study with 12 monthly study points: 6 months before and 6 months after implementation of the CNS-led protocol (Figure 2). Data including laboratory test results, hospitalizations, and mortality documentation were collected from patient records. The nephrologists were integral to the protocol development team and actively participated in the training program. The CNS-led protocol implementation required the nephrologist to refer eligible patients to the authorized nurse. The nephrologist also wrote the initial medical order, specifying the type of EPO, dose, and desired hemoglobin range. Ongoing management and adjustments within the protocol were then carried out by the trained CNSs.

### 2.7. Ethical Considerations

Ethical approval was granted by the institutional review board (IRB #0287-11). Eligible patients received a verbal explanation of the study from the research team, and those who agreed to participate provided written informed consent for participation and publication. Ethical approval was specifically granted for a minimum of 30 patients under this protocol.

### 2.8. Data Collection

Prospective data collection was performed by trained dialysis nurses for participants who met the inclusion criteria and consented to participate. Data included 6 monthly laboratory test results from the period during which the nephrologist managed the HgB balance, followed by 6 months of data during which the CNS assumed an equivalent role. Where possible, data from corresponding months were compared in order to minimize potential seasonal variations in HgB levels.

### 2.9. Parameters Measured

#### 2.9.1. Dialysis Quality Indicators 

Quality indicators were selected based on the 2024 global guidelines for managing chronic kidney failure [44]:

Urea reduction rate (URR): normal value > 65%.

Iron saturation (TSAT): normal range 20–50%.

Infection events: any occurrence of fever, chills, or empiric antibiotic administration which may affect hemoglobin levels and response to erythropoietin.

Bleeding events: any episode requiring blood transfusion or hospitalization, as well as non-acute bleeding (e.g., nosebleeds or minor fistula bleeding).

#### 2.9.2. Process Parameters

Documentation of ESA medical order updates and dosage adjustments was obtained.

#### 2.9.3. Hemoglobin Maintenance Parameters

Average HgB level and changes relative to the previous period were determined, including the hemoglobin variability index [50], which is defined as monitoring HgB levels over a minimum of 6 months to assess stability; low variability was defined as fluctuations of 1–2 g/dL; high variability was defined as fluctuations exceeding 2 g/dL.

#### 2.9.4. Patient Demographics

Data included age, gender, education level, and religious background. Clinical verification of sample criteria was conducted, and ESA type was recorded to ensure that this did not confound study measures.

#### 2.9.5. Perceived Empowerment of CNS

Perceived empowerment was assessed using the 24-item Matthews, Scott, Gallagher, and Corbally questionnaire [51], which was translated from English to Hebrew by two translators using the back–forward method. The measure includes four dimensions: (a) control over professional practice and available resources; (b) perceived support from management and colleagues; (c) recognition by members of the multidisciplinary team including professional autonomy; (d) skills in practice and professional knowledge. The scale ranges from 1–6, where a low score represents a high perception of empowerment. Cronbach’s alpha was 0.928.

## 3. Results

Of the total cohort, 21% (n = 39) were successfully managed under the nurse-led protocol. The demographic characteristics, as presented in Table 2, indicate a balanced gender distribution, a mean age of 68.35 years, and a majority of Jewish and married participants. Twelve nursing staff members completed the protocol training and 10 nephrology CNSs who successfully completed the advanced nephrology course and passed the final examination were formally authorized to manage HgB balance in dialysis patients. The effectiveness of the CSN-led protocol was assessed by dialysis quality indicators, alterations in ESA dosage, consultations with expert nephrologists to update the medical directions, stability of HgB levels, maintaining stability for three months, and the perception of empowerment by CNSs.

### 3.1. Dialysis Quality Indicators

Both iron saturation and urea reduction rate (URR) remained within the established safety thresholds before and after the implementation of the CNS-led protocol (Table 3).

Figure 2 presents the percent of patients who remained within safe values during the periods supervised by nephrologists or CNSs. There were no statistically significant differences in non-acute bleeding incidents between the CNS- and nephrologist-supervised periods (t (32) = 1.614, *p* = 0.12). Notably, there was a significant reduction in the monthly incidence of infectious diseases following the implementation of the nurse-led protocol (t (32) = 2.17, *p* < 0.05), suggesting a potential benefit of the CNS-supervised care in minimizing infection-related complications (Figure 3).

### 3.2. ESA Medical Order Updates and Changes in Dosage

There were no changes made to the type of ESA during the study period. There was a significant decrease in the frequency of consulting an expert nephrologist to update the prescription under CNS supervision (4.61–4.94 to 3.21–3.99 times a month in the periods under nephrologist and CNS supervision, respectively, *p* < 0.05, between the parallel study points 1, 3, and 6, Table 4). This reflects 66 and 54 changes in patient dosage during the periods under nephrologist and CNS supervision, respectively.

### 3.3. HgB Level Stability

The average HgB level in the nephrologist-supervised period before the protocol was implemented was 11.4 g/dL. This was maintained faithfully in the CNS-supervised period (Table 5). Analysis of variance yielded a non-significant result (F(6,128) = 0.462, *p* = 0.83), suggesting that there were no differences in this variable during the study. Similarly, individual comparisons between the corresponding values before and after protocol implementation did not detect any significant differences (43.13% maintained a normal range of HgB compared to 49.25% in the periods supervised by nephrologists and CNSs, respectively). Notably, the percentage of patients who maintained HgB values in the normal range for 3 consecutive months increased after protocol implementation (from 29.7% to 37%, respectively). HgB variability, as shown in Table 6, remained the same throughout the study, although there was a non-significant decrease (χ^2^ (2) = 1.33, *p* = 0.51) in patients with high variability after the implementation of the CNS-led protocol.

The mean absolute HgB values for males decreased from 11.6 to 11.3 g/dL after protocol implementation and increased from 11.2. to 11.5 g/dL for males and females, respectively.

### 3.4. Perceived Empowerment of CNS

Of the 12 nurses, 9 completed the questionnaire for a second time and the pre- and post-intervention answers were compared by a dependent samples t-test for 9 pairs of matched observations. The mean (±SD) scores were 2.02 ± 0.63 before and 1.84 ± 0.75 after implementing the protocol. This difference was not significant—t (8) = 0.8, *p* = 0.45.

## 4. Discussion

This intervention study evaluated the role of nephrology clinical nurse specialists (CNSs) in the individualized and carefully tailored management of HgB levels in hemodialysis patients. Given that HgB variability is associated with increased morbidity, mortality, and the need for blood transfusions [26,52], maintaining stability is critical. The development of the personalized nephrology-CNS-led protocol was designed to maintain stable HgB levels in dialysis patients requiring specialized medical attention, particularly given the limited availability of expert nephrologists. To ensure clinical safety, a rigorous evaluation process was conducted to determine whether dialysis quality indicators and other patient outcomes were maintained or improved following protocol implementation. The results demonstrate that nephrology CNSs effectively maintained HgB levels, comparable to those achieved by expert nephrologists, confirming that the CNS-supervised protocol is as safe as physician-led care.

Additionally, other dialysis quality indicators (e.g., urea reduction ratio (URR), iron saturation), remained within the same range, irrespective of whether the management was led by nephrologists or CNSs. This reflects the safety and stability of the CNS-led protocol. Moreover, the frequency of nephrologist consultations for updating dosages was lower than before the implementation of the protocol, which validates the personalization approach. The percentage of patients with high HgB variability did not change following the implementation of the CNS-led protocol.

The innovation of this protocol lies in the collaboration between the medical and nursing team and the delegation of authority to nephrology CNSs. The provision of highly personalized, patient-centered care is consistent with previous research demonstrating the benefits of CNS-led interventions [53]. This may be expected since their continuous contact with patients on a near-daily basis positions nurses to implement timely interventions, thereby facilitating continuous improvements in quality of care. In addition, such CNSs possess specialized expertise and can be empowered to optimize anemia management for hemodialysis patients within a defined and narrow range.

Our findings highlight the value of nephrology CNSs and their role in enhancing the efficiency, effectiveness, and safety of HgB management in a high-risk patient population. Furthermore, the observed reduction in dosage changes suggests that CNS involvement successfully reduced nephrologist workload without compromising patient outcomes, which was a primary motivation for implementing the CNS-led protocol. Interestingly, the post-intervention increase in the empowerment score, while not significant (probably due to the low number of observations (n = 9)), may suggest that responsibility increases perception of empowerment among nurses.

The rigorous training of nephrology CNSs and the improvement in outcomes for this patient population suggests potential for expanding CNSs’ responsibilities in other areas, a practice already implemented in some settings. Furthermore, given that the protocol implementation adhered to safety standards, it may be a viable option for outpatient treatment centers facing shortages in nephrologists. Future protocol refinement could explore expanding the protocol to include the management of calcium and phosphorus balance. While these are complex issues, CNSs have an in-depth understanding of patients and their position allows frequent contact with patients and enables them to address critical aspects such as phosphate binder adherence, considering potential side effects, particularly in patients with appetite issues, potentially titrating dosages. Furthermore, the comprehensive assessment of iron levels and related factors could also be integrated to enhance impacts on overall care quality.

The integration of nephrology CNSs into community-based healthcare settings has the potential to enhance health equity by improving access to essential services for dialysis patients. It is imperative that policymakers take proactive steps to expand the competencies and skill sets of nurses, enabling them to qualify as nephrology CNSs. Given the aging population and the increasing burden of chronic diseases, establishing nurse-led nephrology clinics should be strategically prioritized to aid us in meeting the specialized needs of dialysis patients. By implementing these measures, healthcare policymakers can recognize and reinforce the indispensable contributions of CNSs in optimizing patient care, reducing the burden on dialysis patients, and mitigating the strain on highly specialized nephrologists [54]. Additionally, ensuring timely and well-coordinated referrals to appropriate healthcare services will be essential for maintaining continuity of care across multiple settings [54]. Like CNSs in other specialties, nephrology CNSs play a crucial role in advancing high-quality patient care and driving systemic improvements within healthcare systems.

To further strengthen the role of CNSs, policymakers should actively promote specialized nursing education, enabling more nurses to achieve CNS certification and fostering interdisciplinary collaboration among healthcare professionals [55]. As life expectancy increases and the prevalence of chronic diseases rises, it remains imperative for CNSs to clearly define their scope of practice and advocate for their role within healthcare organizations, in administration, and among patients.

### 4.1. Study Limitations

Our single-center design, with a pre–post comparison methodology, while appropriate for this intervention, has constrained generalizability to different clinical settings, particularly community-based units with potentially more stable patient populations. The sample size reflects the difficulties of identifying eligible patients within an outpatient hospital unit that has evolved during the study period and now receives patients with more complex conditions. Furthermore, as previously explained, the dropout rate from the protocol was significant. As mentioned previously, finding hemodialysis patients with stable yet specifically ranged HgB values is uncommon in clinical practice, as these levels typically fluctuate significantly due to their underlying morbidities. Nevertheless, although the study sample provides valuable insights into the extended role of CNSs, the size of the sample warrants the repetition of this study with a larger sample. The absence of patient experience measures (satisfaction, quality of life) represents a missed opportunity to evaluate the patient’s perspective on CNS-led versus physician-led care.

### 4.2. Future Studies

Since anemia itself is associated with severity of illness, there is an enhanced need to regulate HgB [56]. It could therefore be worthwhile to expand the inclusion range to allow more patients to benefit from protocol implementation; however, further research will be required to ensure safety. In addition, although the findings of this intervention study provide valuable, meaningful, and significant contributions to the field, it remains essential and necessary to replicate and validate this research in both hospital-based and community-based healthcare settings in order to further strengthen, expand, and substantiate the findings.

## 5. Conclusions

The results of this study demonstrate that nephrology CNSs play an essential role in enhancing the efficiency, effectiveness, and safety of HgB management in ESRD patients who present a high-risk patient population. The personalized nephrology CNS-led protocol maintained average HgB levels comparable to those achieved by expert nephrologists. This validates the protocol as safe as physician-led care, with other dialysis quality indicators remaining within the same range, irrespective of whether the management was led by nephrologists or CNSs. The involvement of CNSs successfully reduced nephrologist workload without compromising patient outcomes, highlighting the value of nephrology CNSs in improving patient care and mitigating the shortage of expert nephrologists. The integration of nephrology CNSs into community-based healthcare settings has the potential to enhance health equity by improving access to essential services for dialysis patients. It is imperative that policymakers take proactive steps to expand the competencies and skill sets of nurses, enabling them to qualify as nephrology CNSs. Given the aging population and the increasing burden of chronic diseases, establishing nurse-led nephrology clinics should be a strategic priority to meet the specialized needs of dialysis patients.

A personalized protocol led by nephrology CNSs for the treatment of dialysis patients with anemia may present a solution for the low availability of expert nephrologists, if applied within defined safety ranges. In addition, a personalized plan may provide a better fit with patient lifestyle, perhaps preventing the complications that are associated with excessive hemoglobin variability. It should be noted that the implementation of such protocols requires continuous effort, which should be considered vis-a-vis the benefits demonstrated.

### A Closing Note

Although we have been successful with the CNS-led HgB management protocol, with the increasing complexity of patient care, it has become increasingly difficult to identify a “stable patient” who has undergone dialysis for three months with no complex medical history requiring the attention of expert nephrologists. We have therefore discontinued the protocol in our division, although we highly recommend its use with defined, pre-set HgB values in ambulatory dialysis units that serve a more stable patient population. Nonetheless, the involvement of nephrology CNSs in balancing the anemia of dialysis patients is of great importance, regardless of the EPO adjustment, and should be acknowledged and encouraged.

## Figures and Tables

**Figure 1 healthcare-13-01317-f001:**
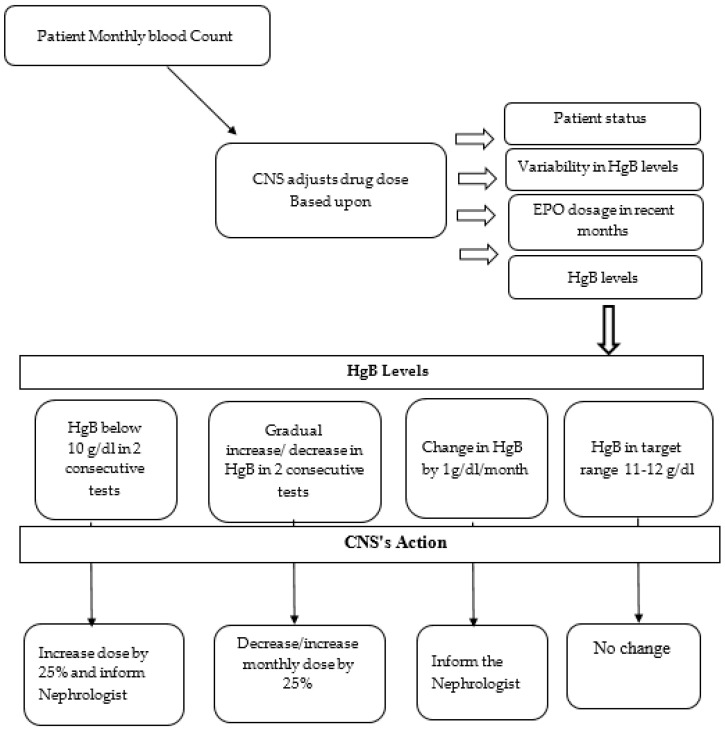
Personalized protocol for HgB balance led by a nephrology CNS.

**Figure 2 healthcare-13-01317-f002:**
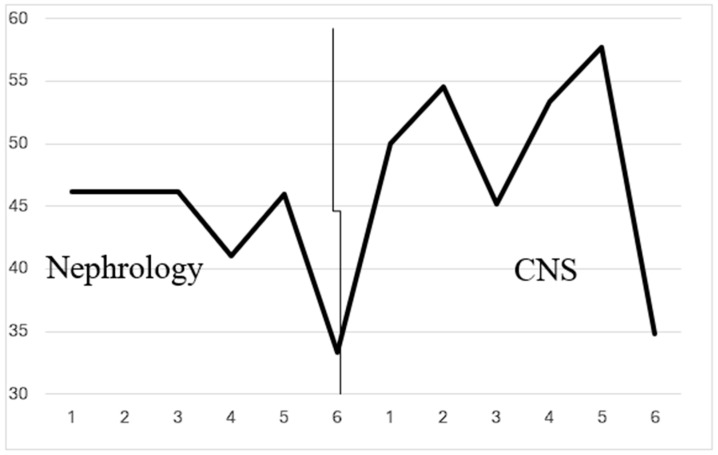
The % of patients who remained within safe values in the nephrology versus the CNS-supervised periods.

**Figure 3 healthcare-13-01317-f003:**
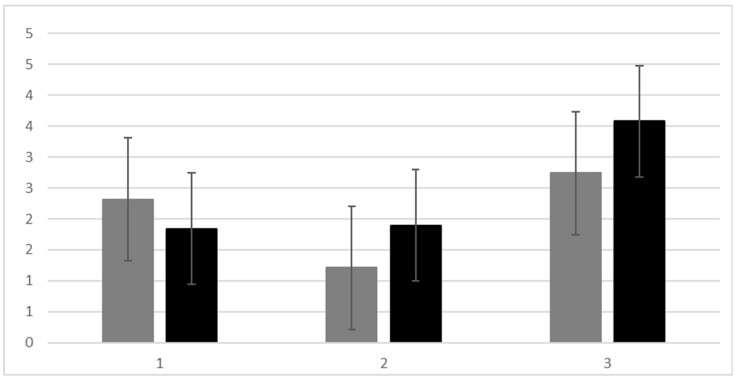
Bleeding and infection incidents. Note: gray—met the target range for 3 consecutive months; black—did not meet the target range for 3 consecutive months. 1—Infectious incidents without hospitalization; 2—non-acute bleeding events; 3—drug dose change.

**Table 1 healthcare-13-01317-t001:** Dropout of hemodialysis patients from the study.

Phase	Phase	Number of Patients’ Dropouts	Updated Number of Participants	% Patients Out of Original Sample	Total Dropouts
Before	1	1	38	97.44	4
	2	0	38	97.44	
	3	0	38	97.44	
	4	0	38	97.44	
	5	2	36	92.31	
	6	1	35	89.74	
Post	1	1	34	87.18	12
	2	1	33	84.62	
	3	2	31	79.49	
	4	1	30	76.92	
	5	4	26	66.67	
	6	3	23	58.97	

**Table 2 healthcare-13-01317-t002:** Patient characteristics.

N = 39	
Age (Average)	68.35 (13.8) years
Sex	53% male
Marital status	Married 79%
	Single 5%
	Divorced 5%
	Widow 11%
Education	
Elementary	25.60
High school	17.90
Higher/Academic	30.8%
Religion	
Jewish	58.9%
Muslim	38.4%
Christian	2%
Place of birth	
Israel	67%
North Africa	10%
Russia	7.60%
East Europe	5%
Background disease (more than 1 disease—3)	
Diabetes Mellitus	23 (58.97%)
Hypertension	5 (12.8%)
Renal cause of ESRD	4 (10.25%)
Heart failure	7 (17.94%)
Vascular disease	2 (5.12%)
Renal cell carcinoma	1 (2.56%)

**Table 3 healthcare-13-01317-t003:** Dialysis quality indicators before and after implementation of CNS-led protocol.

Study Period	Month	Patients	Transferrin Saturation ^1^ (%)	URR ^2^ (%)	Infectious Disease (Average of Events per Month)	Non-Acute Bleeding (Average of Events per Month)
Nephrologist Balance	1	39	(11.68) 33.89	76.77	1.42 events	0.97 events
	2	39		(9.34) 75.79		
	3	39		(9.56) 75.25		
	4	39	(15.28) 31.04	(7.01) 75.67		
	6	37		(7.07) 77.82		
	6	36		(7.77) 76.74		
CNS Balance	1	34	(10.24) 31.18	(8.25) 78.94	0.85 events ^3^	0.61 events
	2	33		(8.15) 75.86		
	3	31		(8.04) 77.85		
	4	30	(16.62) 24.75	(8.84) 77.48		
	5	26		(8.38) 76.9		
	6	23		(7.96) 79		

^1^ Normal Transferrin saturation (according to KDOQI) should be kept between 20% and 50%; ^2^ URR above 65% is recommended for dialysis patients; ^3^ Significant difference *p* < 0.05.

**Table 4 healthcare-13-01317-t004:** ESA medical order update and a change in dosage before and after implementation.

Study Period	Month	Patients	No. Updates of Medical Order (per Month)	Change in ESA Dosage (Number of Patients)
Nephrologist Supervision	1	39	(3.54) 4.74 *	12 (30.8%)
	2	39	(3.84) 4.61	12 (33.3%)
	3	39	(3.93) 4.85 *	15 (38.5%)
	4	39	(3.99) 4.94	8 (20.5%)
	5	37	(3.79) 4.69	8 (21.6%)
	6	36	(3.66) 4.65 *	11 (32.4%)
CNS Supervision	1	34	(3.2) 3.21 *	9 (23.1%)
	2	33	(2.83) 3.42	10 (32.3%)
	3	31	(3.08) 3.32 *	13 (44.8%)
	4	30	(3.55) 3.99	8 (29.6%)
	5	26	(3.26) 3.67	8 (33.3%)
	6	23	(3.23) 3.64 *	6 (28.6%)

* Significant difference between the corresponding study points regulated by nephrologists or nurses.

**Table 5 healthcare-13-01317-t005:** Hemoglobin stability before and after implementation of the CSN-led protocol.

Study Period	Month	Number and (%) of Participants Within Normal Range	Average HgB Levels (SD)	Change in Blood HgB Relative to Previous Period
Nephrologist Balance	1	18 (46.15)	11.29 (1.07)	---
	2	18 (46.15)	11.30 (0.98)	0.013
	3	18 (46.15)	11.42 (0.81)	0.117
	4	16 (41.03)	11.46 (1.02)	0.039
	5	17 (45.95)	11.62 (0.85)	0.161
	6	12 (33.33)	11.52 (0.90)	−0.100
	**Average HgB–11.4 Average % in normal range-43.13**			
CNS Balance	1	17 (50.00)	11.63 (0.84)	0.117
	2	18 (54.55)	11.44 (0.94)	−0.191
	3	14 (45.16)	11.37 (0.78)	−0.078
	4	16 (53.33)	11.27 (1.01)	−0.096
	5	15 (57.69)	11.35 (0.75)	0.078
	6	8 (34.78)	11.50 (0.92)	0.157
	**Average HgB–11.4 Average % in normal range-49.25**			

**Table 6 healthcare-13-01317-t006:** HgB variability before and after protocol implementation.

Study Period	Consistent/Low Variability-Number of Patients (%)	High Variability-Number of Patients (%)	Difference
Nephrologist supervision (n = 39)	53.8 (n = 21)	46.2 (n = 18)	χ^2^(2) = 1.33
CNS Supervision (n = 24)	54.1 (n = 13)	45.8 (n = 11)	

## Data Availability

The data presented in this study are available upon request from R.I (the first author). The data are not publicly available due to privacy reasons.

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
