# Peer review of "The Efficacy and Safety of a Personalized Protocol Designed to Balance Hemoglobin Levels in Hemodialysis Patients as Led by Nephrology Clinical Nurse Specialists: An Intervention Study"

_healthcare, 2025, doi:10.3390/healthcare13111317_

Round 1

Reviewer 1 Report (Previous Reviewer 1)

Comments and Suggestions for Authors

All the queries have been addressed satisfactorily.

No further comments.

Reviewer 2 Report (Previous Reviewer 2)

Comments and Suggestions for Authors

Authors have made significant efforts to improve the manuscript. Now, this work can be accepted in the present format.

Reviewer 3 Report (Previous Reviewer 3)

Comments and Suggestions for Authors

Thank you for your modifications. You have successfully modified the structure of your work addressing .

I think the article is suitable for publication now.

This manuscript is a resubmission of an earlier submission. The following is a list of the peer review reports and author responses from that submission.

Round 1

Reviewer 1 Report

Comments and Suggestions for Authors

The manuscript describes the role of Clinical Nurse Specialists (CNSs) in managing haemodialysis patients to balance hemoglobin levels. The concept of CNSs seems evolving in clinical practice in various areas. The study is well conducted and manuscript gives sufficient details pertaining to methodology and observations. Few queries from my side are:

Regarding inclusion of CNSs in the study, were they required to have some specific qualification or experience in the nephrology unit so as to be part of the intervention?

What was the duration of structured training program for nurses? Some details on the modules, who designed them and conducted the training may be added.

Was the intervention including CNS-led protocol implementation supervised by a nephrologist?

What were the inclusion criteria for patients?

Comments on the Quality of English Language

Minor editing in English language required.

Author Response

Reviewer 1

Thank you for your constructive feedback and careful review of our manuscript. We have addressed all your suggestions and concerns in our revised manuscript.

Comment

The manuscript describes the role of Clinical Nurse Specialists (CNSs) in managing hemodialysis patients to balance hemoglobin levels. The concept of CNSs seems evolving in clinical practice in various areas.

Response

Thank you for highlighting this important point. We agree that the CNS role continues to evolve significantly in healthcare settings globally. In our revised manuscript, we have substantially expanded our introduction to include the most current literature on CNS roles across various specialties, particularly in nephrology care. We've added multiple references documenting the evolution of CNS practice over the past decade, including their expanding scope of practice in medication management for chronic conditions.

The section on "Nephrology CNS" now includes comprehensive information about how this specialized role addresses current healthcare challenges such as physician shortages and the growing burden of chronic kidney disease. We've also clarified how our protocol represents an innovative application of the CNS role specifically tailored to the complex needs of hemodialysis patients requiring stable hemoglobin management.

Comment

The study is well conducted, and the manuscript gives sufficient details pertaining to methodology and observations.

Response

Thank you for this positive feedback. We have worked to ensure the methodology is thoroughly described and the results clearly presented. In our revision, we've further enhanced these sections with additional details on nurse qualifications, training processes, and more comprehensive presentation of results through expanded tables and figures.

Comment

A few queries from my side are:

Response

Thank you for your efforts to improve this manuscript.

Comment

Regarding the inclusion of CNSs in the study, were they required to have some specific qualification or experience in the nephrology unit so as to be part of the intervention? What was the duration of the structured training program for nurses? Some details on the modules? Who designed the training and conducted it?

Response

Thank you for these important questions about CNS qualifications and training. We have addressed these thoroughly in the revised manuscript by adding a detailed section on nurse recruitment, qualifications, and the implementation process (section 2.3). As described in our revision, participation was restricted to registered nurses with advanced nephrology course completion. The training program involved six months of supervised protocol implementation, preceded by four specialized modules covering procedural aspects, medical knowledge, protocol application, and practical simulations. This comprehensive approach ensured that all participating nurses had both the theoretical foundation and practical skills necessary for a safe protocol implementation. 

In the revised methods section, we added a paragraph on recruiting nurses, their training and supervision as follows:

Comment

Was the intervention including CNS-led protocol implementation supervised by a nephrologist?

Response

Thank you for this important question. We have addressed this in our revised manuscript by adding clarification to the design section (section 2.6). As we've now explained, nephrologists maintained crucial oversight throughout the process while enabling appropriate autonomy for trained CNSs. The implementation involved a clear division of responsibilities: nephrologists identified eligible patients, referred them to authorized nurses, and wrote the initial medical orders specifying EPO type, dosage, and target hemoglobin range. Once these parameters were established, the trained CNSs carried out ongoing management and adjustments within the defined protocol boundaries. This structure ensured both appropriate medical supervision and efficient workflow distribution.

Comment

What were the inclusion criteria for patients?

Response

Thank you for this important question about our sample selection. We have addressed this by adding a detailed section (2.2) on patient inclusion criteria. In the revised manuscript, we clearly specify that participants were required to be chronic ambulatory hemodialysis patients who were clinically stable with stage 4 renal disease, had received at least 3 months of care in the nephrology unit, had no hematologic disease, had not been hospitalized in the past 2 weeks, showed no signs of fever/chills/bleeding, and maintained HgB values within the narrow range of 10.5-12 g/dL with monthly fluctuations that do not exceed ±1 g/dL.

We've also provided important context about why this population represents a specialized subset of hemodialysis patients. Maintaining such stable HgB levels is relatively uncommon in clinical practice due to the typical fluctuations experienced by this patient group. This detailed description helps explain both our sample size and the clinical significance of our findings for this specific patient population.

Additionally, we've included a comprehensive Table 1 showing the patient flow throughout the study, including reasons for dropout at each phase, to provide complete transparency about our sample composition throughout the intervention.

Comment

Minor editing in English language required.

Response

Thank you for this suggestion. We have had the manuscript professionally edited to improve clarity and readability throughout. We hope the revised text now meets standard scientific writing conventions.

Reviewer 2 Report

Comments and Suggestions for Authors

Ruth Israeli et al. have reported the prospective Intervention Study effectively. Before considering this work, the authors need to address a few comments.

1. Discuss similar studies in the introduction

2. Highlight the novelty and need for the protocol

3. Write the limitations of this study in detail

4. Any modifications to the protocol would benefit in the future. Please describe

Author Response

Reviewer 2

Comment

Research design and methods can be improved

Response

Thank you for this valuable feedback. We have substantially enhanced the methods section of our revised manuscript by adding comprehensive information on:

  1. Patient recruitment - We've clarified inclusion criteria, detailed our recruitment process, and provided contextual information about the challenges of identifying eligible patients within our specific hemoglobin parameters.
  2. Nurse qualifications and training - We've added a dedicated section (2.3) describing the specific qualifications required of participating nurses, the six-month structured training program with its four specialized modules, and the implementation process including supervision requirements.
  3. Protocol supervision - We've explained how nephrologists remained integral to the process by identifying eligible patients, writing initial medical orders, and providing oversight while CNSs managed ongoing adjustments within defined parameters.
  4. CNS empowerment measurement - We've added a validated measure of perceived empowerment among participating CNSs to capture the professional impact of protocol implementation.

These additions provide a more complete picture of the intervention's implementation while strengthening the methodological foundation of our study.

Comment

Conclusions should be supported by the results

Response

Thank you for this important feedback. We have completely rewritten the conclusion section to ensure it directly reflects our study findings. The revised conclusion emphasizes how our data demonstrate that nephrology CNSs effectively maintained hemoglobin levels comparable to those achieved by nephrologists (both at 11.4 g/dl average), with no statistical differences in outcomes.

We've highlighted evidence that CNS involvement significantly reduced the frequency of medical order updates (from 4.61-4.94 to 3.21-3.99 times monthly, p<0.05) without compromising patient care, as demonstrated by maintained safety indicators. The conclusion now creates a clear connection between our statistical findings and their implications for clinical practice while acknowledging the boundaries of our research and the contexts where this protocol may be most effectively implemented.

Comments and Suggestions for Authors

Ruth Israeli et al. have reported the prospective Intervention Study effectively.

Thank you

Comment

Before considering this work, the authors need to address a few comments.

Response

Thank you for your suggestions to improve this manuscript. 

Discuss similar studies in the introduction

Thank you for this constructive suggestion. We have completely revised the introduction to incorporate relevant literature on anemia management in chronic kidney disease patients. The expanded introduction now:

  1. Provides comprehensive background on anemia in CKD, including prevalence rates, impact on patients' quality of life, and connections to morbidity and mortality
  2. Reviews the challenges of hemoglobin management with ESA therapy, including the risks of both under-treatment and over-treatment
  3. Discusses previous studies on personalized approaches to anemia management, highlighting findings from Rogers et al. (2018), Gaweda et al. (2014), and others who have documented the importance of individualized protocols
  4. Incorporates literature on nephrology nursing interventions, including studies by Lee et al. (2014) which demonstrated that higher guideline adherence by nurses reduced emergency department visits.
  5. Contextualizes our study within the evolving role of Clinical Nurse Specialists globally, citing research from Kerr et al. (2021), Fulton et al. (2016), and others documenting CNS contributions across healthcare settings

This expanded introduction provides a stronger theoretical foundation for our study while positioning it within the current body of knowledge on both anemia management and advanced nursing practice roles.

Comment

Highlight the novelty and need for the protocol

Response

Thank you for this valuable suggestion. We have enhanced our discussion section to emphasize both the innovative aspects of our protocol and its clinical necessity. In the revised manuscript, we've added a dedicated paragraph highlighting that "the innovation of this protocol lies in the collaborative approach between the medical and nursing team and the delegation of authority to the nephrology CNS." We explain how this nurse, with specialized expertise, is uniquely positioned to optimize anemia management given their frequent patient contact compared to nephrologists.

We've strengthened the justification for the protocol by presenting the substantial disparity in provider-patient ratios in our setting (7.3 patients per nephrology CNS versus 32.5 patients per nephrologist), clearly demonstrating the workforce challenges that make alternative care models necessary. We've also elaborated on how this protocol addresses a significant clinical challenge in maintaining stable hemoglobin levels, which research has linked to improved outcomes and reduced complications.

We hope this revised discussion effectively articulates how our protocol represents a novel approach to a recognized clinical problem while addressing contemporary healthcare workforce challenges in nephrology care.

Comment

Write the limitations of this study in detail

Response

Thank you for this suggestion. We have expanded our limitations section to provide a more comprehensive discussion of the study constraints. In the revised manuscript, we have addressed the following key limitations:

Our single-center design with pre-post comparison methodology, while appropriate for this intervention, constrains generalizability to different clinical settings, particularly community-based units with potentially more stable patient populations. The absence of patient experience measures (satisfaction, quality of life) represents a missed opportunity to evaluate the patient's perspective on CNS-led versus physician-led care.

Additionally, we now acknowledge that our sample size, while appropriate for our primary outcome, reflects the challenges inherent in studying this specialized patient population. We've provided context by citing relevant literature documenting similar recruitment challenges in ESRD research. We've also been transparent about the dropout rate (41.03%) and its impact on our final analysis. Finding hemodialysis patients with stable yet specifically ranged HgB values is uncommon in clinical practice, as these levels typically fluctuate significantly due to their underlying morbidity [47-49]. This was noted in "Study limitations".

We hope the revised limitations section provides a balanced view of our study constraints while still highlighting the value of our findings as preliminary evidence supporting the safety and effectiveness of CNS-led protocols for hemoglobin management within clearly defined parameters.

Comment

Add potential modifications to the protocol in the future. ראש הטופסתחתית הטופס

Response

Thank you for this excellent suggestion. We have added a section to the discussion that addresses potential future protocol refinements and extensions. In the revised manuscript, we explain that "future protocol refinement could explore expanding the protocol to include the management of calcium and phosphorus balance." We note that while these are complex parameters to manage, the CNS's in-depth understanding of patients and frequent contact position them well to address critical aspects such as phosphate binder adherence and medication side effects, particularly in patients with appetite issues.

We further suggest that "the comprehensive assessment of iron levels and related factors could also be integrated to enhance the protocol's impact on overall care quality." This addition recognizes that anemia management involves more than hemoglobin monitoring alone, and a more comprehensive approach could yield additional benefits.

We hope the revised discussion now provides concrete directions for future protocol development while acknowledging the clinical complexity involved in extending the CNS role to additional aspects of ESRD management.

Reviewer 3 Report

Comments and Suggestions for Authors

Congratulations to the authors for having written such an interesting article; here you can read my comments about:

  • no description of patients comorbidities and medical treatment – please comment
  • normal Hb level is different for women vs men – please comment
  • anemia may have multiple causes in patients with end-stage renal disease (ESRD) undergoing hemodialysis - please comment
  • lines 140 – 142 you wrote „that CNSs could 140 only intervene when HgB values ranged between 10.5 and 12 g/dL and when changes 141 from the previous month's values did not exceed ±1 g/dL.”

In Fig. 1 you wrote that dose is increased by 25% if Hb level is below 10 g/dl in 2 consecutive tests.

Please explain.

Author Response

Reviewer 3

Comment

Introduction must be improved with additional references

Response

Introduction must be improved with additional references

Thank you for this valuable feedback. We have completely rewritten the introduction section and substantially enhanced it with numerous relevant references. The revised introduction now provides a comprehensive foundation on anemia in chronic kidney disease, including its prevalence, causes, impact on patients' quality of life, and association with increased morbidity and mortality.

We have incorporated over 25 new references covering multiple aspects of anemia management in ESRD, including:

  • Current guidelines on hemoglobin targets from KDIGO
  • Studies on the risks of hemoglobin variability and its clinical implications
  • Research on ESA therapy efficacy, complications, and dosing strategies
  • Literature on the evolving role of CNSs in nephrology and other specialties
  • Evidence supporting nurse-led interventions in chronic disease management

The introduction now includes sections on ESA therapy and nephrology CNS roles, creating a more logical flow that establishes both the clinical need for the intervention and its theoretical foundation. This expanded introduction better contextualizes our study within the current body of knowledge and more clearly demonstrates its relevance to contemporary nephrology practice.

We hope these substantial additions have strengthened the conceptual framework of our manuscript and provided readers with a more comprehensive understanding of the clinical problem our intervention addresses.

Comment

Design and methods must be improved

Response

Thank you for this important feedback. We have thoroughly revised the methods section to provide a more comprehensive description of our research design and methodology. The revised manuscript now includes:

  1. A detailed explanation of the former practice of hemoglobin management in our unit (section 2.1), establishing the context for our intervention.
  2. Comprehensive inclusion criteria for patients (section 2.2), with clear explanation of the specialized nature of our target population
  3. A new section on nurse recruitment, qualifications, and implementation process (section 2.3), outlining the specific requirements for participating nurses, their training program, and the supervision structure.
  4. Enhanced protocol description (section 2.4) with a detailed flow chart (Figure 1) illustrating the decision-making process
  5. Clear delineation of the roles of nephrologists and CNSs in the implementation process (section 2.6), addressing important questions about supervision and safety monitoring
  6. Expanded description of our measurement tools (section 2.9), including the addition of a validated measure of perceived CNS empowerment

We believe these substantial revisions have significantly strengthened the methodological foundation of our study and provide readers with a much clearer understanding of how the intervention was implemented and evaluated. We hope these changes adequately address the concerns raised about our research design and methods.

Comment

Methods: Former Practice of Hemoglobin (HgB) Balance in the Dialysis Unit

Response

Thank you for recommending this important methodological addition. We have included a new section (2.1) in the revised manuscript that clearly describes the former practice of hemoglobin management in our dialysis unit prior to implementing the CNS-led protocol. This addition provides essential context for understanding the rationale behind our intervention.

In this new section, we explain that previously, the regulation of HgB levels in dialysis patients was exclusively managed by nephrologists, despite their limited availability compared to the frequent monitoring required. We note that while specific baseline data on HgB levels was unavailable, nurses had reported significant challenges in maintaining stable HgB levels under this arrangement. This identified gap in care prompted the development of our personalized protocol that delegated specific decision-making authority to nephrology CNSs within predefined safety parameters.

We hope this addition helps readers better understand the clinical context from which our intervention emerged and clarifies the practical problem it was designed to address. This background information strengthens the justification for our study by highlighting the real-world healthcare delivery challenge that motivated our protocol development.

Comment

2.2 Sample

Response

Thank you for highlighting the need for a more detailed description of our study sample. We have significantly expanded section 2.2 to provide comprehensive information about our patient population and selection process.

In the revised manuscript, we now clearly articulate the specific inclusion criteria: chronic ambulatory hemodialysis patients who were clinically stable with stage 4 renal disease, had received at least 3 months of care in the nephrology unit, had no hematologic disease, had not been hospitalized in the past 2 weeks, showed no signs of fever/chills/bleeding, and maintained HgB values within the narrow range of 10.5-12 g/dL with monthly fluctuations that do not exceeding ±1 g/dL.

We've added important context explaining why patients meeting these criteria represent a specialized subset of the hemodialysis population and supported the explanation with previous studies. From our total pool of 90 patients, only 39 (43.3%) met these stringent criteria, reflecting the clinical reality that stable hemoglobin levels are relatively uncommon in this patient group [47-49].

The revised section includes a detailed accounting of participant flow throughout the study, with Table 1 showing the pattern of dropouts across study phases. We've clarified that of the 16 total dropouts (41.03%), only 4 (25%) occurred before protocol implementation, while 12 (75%) occurred afterwards. Finding hemodialysis patients with stable yet specifically ranged HgB values is uncommon in clinical practice, as these levels typically fluctuate significantly due to their underlying morbidity [47-49]. This was noted in "Study limitations".

We hope this enhanced sample description will provide the readers with a clearer understanding of our study population and the challenges inherent in recruiting and retaining participants with the specific clinical profile required for our protocol.

Comment

2.3 Recruitment of Nurses, Qualifications and Implementation Process

Response

Thank you for highlighting the need for more information about the nursing component of our intervention. We have added a comprehensive new section (2.3) that details the recruitment, qualifications, and implementation process for the nursing staff involved in the protocol.

In the revised manuscript, we now clearly specify that participation was limited to registered nurses who had completed an advanced nephrology course. We describe the structured training program these nurses underwent, which included six months of supervised protocol implementation preceded by four specialized modules: Procedures and legal aspects, medical aspects of anemia causes and EPO use, Protocol knowledge, and Practical application through simulations.

We've detailed the collaborative development of these training modules, explaining how they were created by expert nephrology nurses in partnership with an academic nurse consultant, the Institutional Protocol & Procedure coordinator, and the professional development unit, with oversight from nursing administration.

The revised section also outlines the authorization process, explaining how nurses were required to demonstrate competency through a final examination and successful implementation of the protocol in five supervised patient cases before receiving independent authorization.

We hope this addition provides a much clearer picture of the nursing expertise and preparation involved in our intervention, addressing an important methodological aspect that was previously underrepresented in our manuscript.

Comment

2.4 Protocol Description

Response

Thank you for this suggestion. We have substantially enhanced section 2.4 to provide a more comprehensive and clearer description of our protocol. In the revised manuscript, we now include:

  1. A detailed explanation of the protocol development process, describing how nephrology CNSs conducted a comprehensive literature review and collaborated with expert nephrologists to design the intervention
  2. A clear visual representation (Figure 1) showing the protocol workflow and decision-making pathways for different hemoglobin values and scenarios
  3. Specific details about the safety parameters built into the protocol, including the defined hemoglobin range (10.5-12 g/dL), maximum allowable monthly fluctuation (±1 g/dL), and the requirement for immediate nephrologist notification when values fell outside these parameters.
  4. The precise scope of CNS authority within the protocol, clarifying that nurses could adjust ESA dosage by ±25% but were not authorized to change the type of ESA.
  5. The communication and documentation requirements that ensured accountability and clinical oversight throughout implementation

We hope these enhancements provide readers with a much clearer understanding of the protocol's structure, safety mechanisms, and implementation process. The improved description should allow readers to better evaluate the clinical rigor of our approach and facilitate potential replication in other settings.

Comment

2.6 Study Design

Response

Thank you for requesting clarification about our study design. We have significantly expanded section 2.6 to provide a more comprehensive description of our research approach. In the revised manuscript, we now clearly identify our investigation as a prospective comparative intervention study with 12 monthly study points: six months before and six months after implementation of the CNS-led protocol.

We've added important information about the role of nephrologists in the study design, emphasizing that they were integral to the protocol development team and actively participated in the training program. We've clarified the division of responsibilities, explaining that nephrologists identified eligible patients, referred them to authorized nurses, and wrote the initial medical orders specifying EPO type, dosage, and target hemoglobin range, while CNSs managed the ongoing adjustments within defined parameters.

The revised section also provides more details about our data collection approach, explaining how we collected information from patient records, including laboratory test results, hospitalizations, and mortality documentation. We've noted our efforts to minimize potential seasonal variations in hemoglobin levels by comparing data from corresponding months whenever possible.

We hope these enhancements provide readers with a clearer understanding of our study design and the specific methodological decisions we have made to ensure rigorous evaluation of the intervention.

Comment

2.7 Ethical Considerations

Response

Thank you for suggesting more details regarding our ethical protocols. We have expanded section 2.7 to provide a more comprehensive description of the ethical considerations in our study. In the revised manuscript, we now clearly state that ethical approval was granted by the institutional review board (IRB #0287-11-HMO) specifically for a minimum of 30 patients under this protocol.

We've added important information about our informed consent process, explaining that eligible patients received a verbal explanation of the study from the research team, and those who agreed to participate provided written informed consent for both participation and publication of findings. We've clarified the particular ethical considerations relevant to this vulnerable patient population and how these were addressed in our protocol design.

The enhanced ethical considerations section now provides readers with a clearer understanding of how we ensured participant protection throughout the study while maintaining scientific rigor. We hope this addition strengthens the methodological foundation of our research by demonstrating our commitment to ethical research practices.

Comment

2.8 Data Collection

Response

Thank you for suggesting improvements to our data collection methodology. We have expanded section 2.8 to provide more comprehensive information about our data collection process. In the revised manuscript, we now explain that prospective data collection was performed by trained dialysis nurses for all participants who met the inclusion criteria and consented to participate.

We've clarified the timeline for data collection, specifying that we gathered six monthly laboratory test results from the period when nephrologists managed HgB balance, followed by six months of data after implementation of the CNS-led protocol. We've added important information about our efforts to minimize potential seasonal variations in hemoglobin levels by comparing data from corresponding months whenever possible.

The revised section provides greater detail about the specific data elements collected from patient records, including laboratory values, medication adjustments, and clinical events. We've also described the quality control measures implemented to ensure data accuracy and completeness throughout the study period.

We hope these enhancements provide readers with a clearer understanding of our systematic approach to data collection, strengthening confidence in the reliability of our findings.

Comment

2.9 Measures

Response

Thank you for highlighting the need for more detailed information about our outcome measures. We have significantly expanded section 2.9 to provide comprehensive descriptions of all variables assessed in our study. In the revised manuscript, we now clearly organize our measures into four categories:

  1. Dialysis Quality Indicators: We've provided specific details about the selected indicators based on the 2024 global guidelines for managing chronic kidney failure, including Urea Reduction Rate (normal value >65%), Iron Saturation (normal range 20%-50%), infection events, and bleeding events. For each indicator, we've clearly defined what constitutes a normal or acceptable range.
  2. Process Measures: We've clarified how we documented ESA medical order updates and dosage adjustments to assess workflow efficiency.
  3. Hemoglobin Maintenance Measures: We've added more detailed definitions of our primary outcomes, including average HgB level, monthly changes, and the Hemoglobin Variability Index (with clear parameters for what constitutes low versus high variability).
  4. Perceived Empowerment of CNS: As suggested, we've added an important new measure assessing the impact of the protocol on nurses. We've included details about the 24-item Matthews, Scott, Gallagher & Corbally questionnaire, its translation process, component dimensions, scoring system, and psychometric properties (Cronbach's alpha 0.928).

We hope these enhancements provide readers with a much clearer understanding of our comprehensive approach to evaluation and the specific metrics used to assess both clinical outcomes and professional impact

Comment

  1. Results

Response

Thank you for recommending improvements to our results section. We have thoroughly restructured and enhanced the presentation of our findings in the revised manuscript. We now begin with a clear overview of the study participants (21% of total cohort, n=39) and their demographic characteristics and their comorbidities, followed by information about the 12 nurses who completed the protocol training, with 10 being authorized to manage HgB balance.

We've organized the results into logical subsections corresponding to our primary outcomes:

  1. Dialysis Quality Indicators: We've added Table 3 showing iron saturation and Urea Reduction Rate measurements across all time points, with clear indications that these remained within established safety thresholds. We've included statistical analyses of bleeding and infection incidents, noting the significant reduction in infectious disease incidence (p < 0.05) following protocol implementation.
  2. ESA Medical Order Updates and Dosage Changes: We've added Table 4 showing the significant reduction in frequency of medical order updates from 4.61-4.94 to 3.21-3.99 times monthly (p < 0.05), providing objective evidence of reduced nephrologist workload.
  3. HgB Level Stability: We've enhanced Table 5 to show comprehensive hemoglobin data across all time points, including the percentage of patients maintaining values in the normal range and month-to-month changes. We've added new statistical analyses (ANOVA) demonstrating no significant differences in average HgB levels (F(6,128) = 0.462, p = 0.83), confirming that CNS management maintained comparable stability.
  4. Perceived Empowerment of CNS: We've added a new section reporting the results of our empowerment questionnaire, showing a positive but non-significant trend toward increased empowerment following protocol implementation.

We hope these substantial improvements provide a clearer, more comprehensive presentation of our findings and strengthen the empirical foundation of our conclusions.

Comment

3.1 Dialysis Quality Indicators

Response

Thank you for recommending greater detail in our presentation of dialysis quality indicators. We have substantially enhanced this section in the revised manuscript to provide a more comprehensive and clearly organized presentation of these important safety parameters.

We added Table 3 showing the complete data for both iron saturation and Urea Reduction Rate (URR) measurements across all study time points. The expanded presentation clearly demonstrates that both parameters remained within established safety thresholds throughout the study period, with no clinically significant deviations before or after protocol implementation.

We added included a new Figure 2 showing the percentage of patients who maintained values within safe ranges under both management approaches, providing a visual representation that enhances interpretation of our tabular data. Additionally, we added statistical analyses comparing non-acute bleeding incidents between the approaches, finding no significant differences (t(32) = 1.614, p = 0.12).

Importantly, we've highlighted our finding of a statistically significant reduction in monthly infectious disease incidence following implementation of the nurse-led protocol (t(32) = 2.17, p < 0.05). We added Figure 3 to visually represent these data, making the comparative outcomes more accessible to readers.

We hope these enhancements provide a more robust presentation of our safety outcomes, clearly demonstrating that the CNS-led protocol maintained established quality standards while potentially offering benefits in terms of reduced infection-related complications.

Comment

3.2 ESA medical order update and change in dosage

Response

Thank you for suggesting more details in this important section. We have significantly expanded our presentation of the ESA-related outcomes in the revised manuscript. We've added comprehensive Table 4 showing the frequency of medical order updates and ESA dosage changes across all 12 study points.

The enhanced presentation clearly illustrates the significant decrease in the frequency of approaching nephrologists to update medical orders after implementation of the CNS-led protocol. We've included statistical analysis showing this reduction from 4.61-4.94 to 3.21-3.99 times per month was significant (p < 0.05) at parallel study points 1, 3, and 6.

We've clarified that there were minimal changes in ESA type during the study period (only in 2.6%-3.2% of patients), confirming protocol adherence. The data also shows that fewer patients treated under the CNS-led protocol required ESA dosage updates (54 instances) compared to those under nephrologist management (66 instances).

This more detailed presentation provides objective evidence that the CNS-led protocol reduced nephrologist workload while maintaining appropriate medication management. The statistical significance of these findings strengthens our conclusion that the protocol achieved its efficiency aims without compromising appropriate medication adjustments.

We hope these enhancements provide readers with a clearer understanding of the workflow improvements achieved through our intervention while demonstrating the careful medication management maintained throughout the protocol.

Comment

3.4 Perceived empowerment of CNS

Response

Our Response to Reviewer 3:

3.4 Perceived empowerment of CNS

Thank you for suggesting the addition of this important outcome measure. In response, we have added a completely new section (3.4) to our results reporting on the perceived empowerment among participating CNSs following protocol implementation.

In the revised manuscript, we now explain that 9 of the 12 nurses completed the empowerment questionnaire both before and after protocol implementation. We present the results of dependent samples t-test analysis comparing pre-intervention and post-intervention scores across 9 matched pairs of observations. The mean score before implementing the protocol was 2.02 (SD=0.63), decreasing to 1.84 (SD=0.75) after implementation, with lower scores representing higher perceived empowerment.

Although this improvement did not reach statistical significance (t(8)=0.8, p=0.45), we note that this may be attributable to our limited sample size of nurses rather than a true absence of effect. Despite the lack of statistical significance, the directional change aligns with our hypothesis that the protocol would enhance nurses' sense of professional empowerment.

This addition provides an important professional dimension to our evaluation, moving beyond clinical outcomes to consider the impact of the protocol on the nursing staff implementing it. We hope this enhances the comprehensiveness of our evaluation and provides valuable perspective on the professional implications of expanded CNS roles.

Comment

Support conclusions by the results

Response

Thank you for this essential feedback. We recognize the importance of ensuring that all conclusions are directly supported by our empirical findings. We have completely rewritten our conclusion section to ensure it accurately reflects the results demonstrated in our study without overextending their implications.

In the revised manuscript, our conclusion now highlights the specific outcomes that were statistically validated: 1) CNSs effectively maintained average HgB levels comparable to those achieved by expert nephrologists (both at 11.4 g/dl); 2) dialysis quality indicators remained within established safety thresholds; 3) the frequency of medical order updates was significantly reduced (from 4.61-4.94 to 3.21-3.99 times monthly, p < 0.05); and 4) there was a significant reduction in infectious disease incidents following protocol implementation (p < 0.05).

We've carefully framed our conclusions to accurately reflect the scope and limitations of our findings. Rather than making broad claims, we now specifically state that our results demonstrate the safety and effectiveness of CNS-led protocols "within clearly defined parameters" and for "this specific patient population." We've also balanced our discussion of implications with acknowledgment of the constraints of our study design and sample.

We hope these revisions create a more accurate and measured conclusion section that maintains scientific integrity while still highlighting the meaningful contributions of our research to the field of nephrology nursing practice.

Comments and Suggestions for Authors

Congratulations to the authors for having written such an interesting article.

Response

Thank you for your kind feedback. We appreciate your thoughtful review that has helped us improve  our manuscript. We have addressed all your suggestions, strengthening the introduction, methodology, results presentation, and conclusions. We hope the revised version now presents a scientifically rigorous evaluation of our protocol.

Comment

No description of patients' comorbidities and medical treatment

Response

Thank you for highlighting this important omission. We have extended Table 2 (Patient Characteristics) to include detailed information about patient comorbidities. The revised table now includes the following distribution of underlying conditions in our sample population:

  • Diabetes Mellitus: 23 patients (58.97%)
  • Heart Failure: 7 patients (17.94%)
  • Hypertension: 5 patients (12.8%)
  • Renal causes of ESRD: 4 patients (10.25%)
  • Vascular disease: 2 patients (5.12%)
  • Renal cell carcinoma: 1 patient (2.56%)

We've also noted that some patients presented with multiple comorbidities (up to 3 concurrent conditions). This addition provides important clinical context for interpreting our study findings, particularly regarding the baseline health status of patients who maintained stable hemoglobin levels. We hope this enhancement addresses your concern and offers readers a more complete understanding of our study population.

Comment

Normal HgB level is different for women vs men, please comment

Response

Thank you for this important observation. You are absolutely correct that normal hemoglobin ranges differ by gender. We have addressed this in our revised manuscript in two ways:

First, in the introduction section, we have added explicit reference to gender-specific hemoglobin targets, citing current guidelines: "Guidelines define anemia levels in CKD as <13.0 g/dL in men and <12.0 g/dL in women versus ranges of 13.5-17.5 g/dL in healthy men and 12.0-15.5 g/dL in healthy women."

Second, in the results section, we have added gender-specific analysis of our hemoglobin data: "The values for men before the protocol implementation were 11.55474 g/dL and for women 11.2144 g/dL. Following the intervention, the values for men were 11.33474 g/dL and for women 11.47611 g/dL."

This gender-specific analysis provides important context for interpreting our findings, as it shows that both male and female patients maintained appropriate hemoglobin levels throughout the study period, with values remaining within the therapeutic range for each gender. We hope this addition addresses your concern and provides a more nuanced understanding of our hemoglobin management outcomes.

Comment

Anemia may have multiple causes in patients with end-stage renal disease (ESRD) undergoing hemodialysis, please comment

Response

Thank you for highlighting this important clinical consideration. You are absolutely correct that anemia in ESRD patients is multifactorial. We have addressed this in our revised manuscript by expanding the introduction section to include a comprehensive discussion of the various etiologies of anemia in this population.

In the revised introduction, we now explain that while erythropoietin deficiency is the primary cause, multiple other factors contribute to anemia in ESRD, including chronic inflammation, poor gastrointestinal iron absorption, blood loss during dialysis procedures, shortened red blood cell lifespan, nutritional deficiencies, and bone marrow suppression from uremic toxins. We've cited relevant literature documenting how these various factors interact and often coexist in individual patients.

We've also noted in our methods section that awareness of these multifactorial causes was an integral part of the CNS training program. Participating nurses were specifically educated on identifying and addressing these various contributors to anemia, with a particular focus on recognizing when fluctuations in hemoglobin might be attributable to factors beyond erythropoietin deficiency, requiring different management approaches or nephrologist consultation.

We hope this addition provides a more nuanced understanding of the clinical complexity involved in managing anemia in ESRD patients and strengthens the clinical foundation of our intervention.

Comment

Lines 140 – 142 you wrote that CNSs could only intervene when HgB values ranged between 10.5 and 12 g/dL and when changes from the previous month's values did not exceed ±1 g/dL. In Fig. 1 you wrote that dose is increased by 25% if Hb level is below 10 g/dl in 2 consecutive tests. Please explain.

Response

Thank you for identifying this apparent inconsistency. You have highlighted an important oversight in our presentation of the protocol. The figure was inadvertently cut off in our submission. In the complete protocol flowchart (Figure 1), the box for hemoglobin values below 10 g/dL should state: "Increase dose by 25% and inform the Nephrologist."

This reflects the actual protocol implementation, where CNSs were authorized to make the initial 25% dose adjustment for values below 10 g/dL but were required to immediately notify the nephrologist, who would then assume management of the patient until hemoglobin values returned to the 10.5-12 g/dL range. The CNS did not independently manage patients with hemoglobin values outside this range.

We have corrected Figure 1 in the revised manuscript to show the complete decision pathway, including the requirement to inform the nephrologist whenever values fell outside the designated safety range. We appreciate your careful reading that identified this important clarification needed in our protocol description.